# Graph Neural Network-Guided Contrastive Learning for Sequential Recommendation

**DOI:** 10.3390/s23125572

**Published:** 2023-06-14

**Authors:** Xing-Yao Yang, Feng Xu, Jiong Yu, Zi-Yang Li, Dong-Xiao Wang

**Affiliations:** School of Software, Xinjiang University, 666, Shengli Road, Urumqi 830049, China; 107552001395@stu.xju.edu.cn (F.X.); yujiong@xju.edu.cn (J.Y.); liziyang@xju.edu.cn (Z.-Y.L.); goldenpool@xju.edu.cn (D.-X.W.)

**Keywords:** sequential recommendation, graph neural networks, contrastive learning, graph neural networks guided

## Abstract

Sequential recommendation uses contrastive learning to randomly augment user sequences and alleviate the data sparsity problem. However, there is no guarantee that the augmented positive or negative views remain semantically similar. To address this issue, we propose graph neural network-guided contrastive learning for sequential recommendation (GC4SRec). The guided process employs graph neural networks to obtain user embeddings, an encoder to determine the importance score of each item, and various data augmentation methods to construct a contrast view based on the importance score. Experimental validation is conducted on three publicly available datasets, and the experimental results demonstrate that GC4SRec improves the hit rate and normalized discounted cumulative gain metrics by 1.4% and 1.7%, respectively. The model can enhance recommendation performance and mitigate the data sparsity problem.

## 1. Introduction

Recommender systems play a key role in e-commerce, video streaming media, music platforms, and many other online services, as they address the problem of information overload. Research on serialized recommendations is of great importance and has gained widespread attention in recent years. Given a sequence of user behavior, sequential recommendations capture the sequential transition pattern between consecutive items and predict the next item that the user may be interested in. GRU4Rec [1] uses recurrent neural networks (RNNs) to model the sequential behavior of users. SASRec [2] uses a self-attention mechanism to capture high-level information from user behavior. Additionally, the recommendation algorithm based on graph neural networks (GNN) converts each behavior sequence into a sequential graph.

Self-supervised learning (SSL) mines self-supervised signals from the sequence to mitigate the problem of data sparsity. Contrastive learning, as a typical self-supervised learning method, has received increasing attention in various fields. Contrastive learning takes positive and negative samples from data to maximize the consistency of positive samples and minimize the consistency of negative samples. S3Rec [3] and CL4Rec [4] perform data-level augmentations to user behavior sequences, while DuoRec [5] performs model-level augmentations.

The above contrastive learning methods completely obtain self-supervised signals from the sequence. Due to the limited items contained in each user behavior sequence, the self-supervised signals obtained from the sequence are insufficient. Moreover, S3Rec and CL4Rec generate contrastive views through simple data augmentation methods (such as item crop) on user-item interaction sequences, resulting in low diversity of contrastive views and weak self-supervised signals. To address the above issues, the main work of this paper is as follows:(1)We use graph neural networks to capture local context information, and an encoder is used to extract sequence context information to generate importance scores.(2)We ensure that the positive view remains semantically similar by guiding the generation of contrastive learning positive views through graph neural networks.(3)Using two data augmentation methods, mask and crop, positive views are generated based on the importance score of the item.

## 2. Related Work

### 2.1. Sequential Recommendation

Sequential recommendation predicts the next item that users may purchase by learning the user’s behavioral sequence. Early research focused on modeling the relationships between items through Markov chains (MC), which are based on the previous item purchased by the user to infer the next item to be purchased. For instance, FPMC [6] captures sequential patterns through first-order Markov chains and then extends to high-order Markov chains. With the development of deep neural networks, the focus of sequential recommendation research has shifted towards the use of neural networks, such as RNN-based [7,8] methods that treat user behavior sequences as a sequential modeling problem and apply recurrent neural networks to capture sequential transformation patterns. CNN-based [9] algorithms treat the sequence as an image and use convolution networks to model sequences.

Inspired by the effectiveness of the self-attention mechanism in the NLP field, SASRec applies the attention mechanism to sequential recommendation for the first time and learns the transfer mode of items by stacking multi-head attention blocks. MIND [10] utilizes dynamic routing to obtain multiple interests of users. GNN-based [11,12,13] models are used to capture structural information in behavior sequences. GC-SAN [14] captures rich local dependencies through the graph neural network and learns long-term dependencies for each sequence through the self-attention mechanism.

### 2.2. Contrastive Learning

The main idea of contrastive learning is to learn information representation through contrasting positive and negative views. This approach has led to significant achievements in computer vision [15], natural language processing [16], and graph neural networks [17]. Recently, some studies have introduced contrastive learning into recommendation systems [18]. For example, SGL [19] takes node self-discrimination as a supervised task, providing auxiliary information for existing GNN-based recommendation models. SEPT [20] designs a socially aware self-supervised framework for learning self-supervised signals from user-item bidirectional graphs and social network graphs. S3Rec designs four auxiliary self-supervised objectives through maximum mutual information for data representation learning. CL4Rec applies three kinds of data augmentation (i.e., crop, mask, and reorder) to generate positive examples. DuoRec proposes a model-level augmentation method that adopts a supervised positive sampling strategy to capture self-supervised signals in the sequence.

As shown in Figure 1, we use two augmentation methods: mask and crop. The item masking strategy randomly masks items and replaces them with a special token [mask]. The idea behind this method is that a user’s intent is relatively stable over a period of time. Therefore, even though some items are masked, the main intended information is still preserved in the remaining items. Random crop is a common data augmentation technique to increase the variety of images in computer vision. It usually creates a random subset of an original image to help the model generalize better. Inspired by the random crop technique in images, we propose the item crop augmentation method for the contrastive learning task in the sequential recommendation.

### 2.3. Guided Method

Recent research has found that data-driven recommendation systems may pose serious threats to users and society, such as spreading fake news on social media to manipulate public opinion or inferring privacy information from recommendation results. Therefore, to mitigate the negative impacts of recommendation systems and increase public trust in the technology, the trustworthiness of recommendation systems is receiving increasing attention. There are already some studies on explainable recommendation systems [21], but these methods mainly explain why the algorithm recommends certain items. GC4SRec focuses on general explanation methods originally designed to identify explanatory feature results [22,23]. By applying these methods to sequential recommendation approaches, it is possible to identify historical items in user sequences that explain the predicted next item and assign importance scores to these historical items. For example, saliency [23] obtains attribution scores of input features by returning gradients relative to the input features. Attention-based mechanisms model items’ relative importance through attention weights. However, using attention as an explanation is controversial.

## 3. GC4SRec

The sequential recommendation model GC4SRec proposed in this paper, with the structure shown in Figure 2, consists of two phases: the graph neural network-guided phase and the contrastive learning phase. In the graph neural network-guided phase, the model transforms the user sequence into a graph and then generates the importance score of each item in the sequence by learning contextual information about the user sequence through GNN. In the sequence contrastive learning phase, the model performs data augmentation methods according to the importance scores to generate positive views.

### 3.1. Graph Neural Network-Guided Approach 

Sequential recommendations predict what users want to click or buy next based on their behavior sequence. Let U be the set of users and V be the set of items. For each user’s behavior sequence, a series of click actions of the user is represented as $S=\{s_{1},s_{2},\ldots \ldots ,s_{n}\}$, where $s_{t}$ denotes the set of items clicked by user u at time t.

The user behavior sequences can be constructed as a sequence graph, and given a sequence $s=(v_{1},v_{2},v_{3},v_{2},v_{4})$, $v_{i}\in V$, where each item $v_{i}$ is considered as a node of the graph, and $\left(v_{1},v_{2}\right)$ is an edge in the sequence graph, indicating that the user clicked on item $v_{1}$ and then clicks on item $v_{2}$ in sequence s. Since several items may appear in the sequence repeatedly, we assign each edge a normalized weight, which is calculated as the occurrence of the edge divided by the outdegree of that edge’s start node. $M^{I},\;M^{O}\in R^{n\times n}$ denotes the weighted connection of the incoming and outgoing edges of the session graph, and the corresponding graph and matrix for sequence s are shown in Figure 3.

GNN is well suited for sequential recommendation because it can automatically extract the features of the sequential graph, considering rich node connections. The node vector v∈ℝd denotes the embedding vector, and the information propagation between different nodes can be formalized as:(1)at=concat(MtI((v1,…,vn)WI+bI), MtO((v1,…,vn)WO+bO)
where WI, WO are the parameter matrices, bI, bO are the bias vectors, MtI, MtO are the *t*-row corresponding to node, vt, and at extracts the contextual information of the neighborhood of vt. Then, we feed it and vt−1 into the GNN. The final output of the GNN layer is calculated as follows:(2)zt=σ(Wzat+Pzvt−1)
(3)rt=σ(Wrat+Prvt−1)
(4)ht′=tanh(What+Ph(rt⊙vt−1))
(5)ht=(1−zt)⊙vt−1+zt⊙ht′
where Wz, Wr, Wh∈ℝ2d×d,Pz, Pr, Ph∈ℝd×d are the trainable parameters, σ(•) is the sigmoid function, and e is the element-wise multiplication. zt, rt are the update gate and reset gate, respectively, deciding what information is to be preserved and discarded. After feeding the sequence to the GNN, the embedding vector H=[h1,h2,…,hn] of all nodes in the sequential graph can be obtained. It is then fed into the self-attention layer to capture the global user preference.
(6)F=softmax((HWQ)(HWK)Td)(HWV)
where WQ, WK, WV∈ℝ2d×d are the projection matrices. On this basis, we apply a two-layer feed-forward network to give the model non-linearity, and a residual connection is added after the feed-forward network to make it easier for the model to leverage the low-layer information.
(7)E=ReLU(FW1+b1)W2+b2+F
where W1, W2 are d×d matrices and b1, b2 are d-dimensional vectors. The above self-attention mechanism is defined as follows:(8)E=Attention(H)
(9)Ek=Attention(Ek−1)

$E^{k}$ is the output of the multi-layer self-attention network. Finally, the next click probability of each candidate item $v_{i}\in s$ is predicted, $y_{i}$ denotes the recommendation probability of the item $v_{i}$, and $v_{i}$ is the item embedding of $v_{i}$. The formula is defined as follows:(10)$$y_{i}=soft\max(E^{k}v_{i})$$

To obtain the importance score of each item in the user sequence, the importance score of each item in sequence s is set to $score(s)=[score(v_{1}),score(v_{3}),score(v_{2}),score(v_{4}),score(v_{3}))$, $score(v_{i})$ denotes the importance score of the item $v_{i}$. In the item embedding matrix $E^{k}$, where $e_{v_{i}}$ is an embedding of item $v_{i}$, the j dimensional importance score of $e_{v_{i}}$ is defined as: $score(e_{v_{i,j}})=\|\frac{\partial y_{i}}{\partial v_{i,j}}\|$. By adding and normalizing the d-dimensional importance score, the importance score of $e_{v_{i}}$ can be obtained as:(11)$$score(v_{i})=\frac{\sum_{j=1}^{d}score(e_{v_{i,j}})}{\sum_{i^{\prime }=1}^{n}\sum_{j=1}^{d}score(e_{v_{i^{\prime },j}})}$$

The value returned for $score(v_{i})$ is between [0, 1], indicating the importance of $v_{i}$ in session s. Importance scores are relative, with all importance scores in a session adding up to 1.

### 3.2. Contrastive Learning

The model uses two augmentation methods: mask and crop. In the crop operation, the model removes some items with low importance scores. Assuming that in the session s, the model removes the lowest k items in terms of scores, where k is defined as |μ⋅|s||,0<μ≤1, and $\left(v_{i_{1}},v_{i_{2}},\ldots ,v_{i_{k}}\right)$ represents the subsequence composed of the lowest k items in the session s. We can define the positive and negative views of the guidance as follows:(12)$$s^{crop^{+}}=s-[v_{i_{1}},v_{i_{2}},\ldots ,v_{i_{k}}]$$
(13)$$s^{{crop}^{-}}=[v_{i_{1}},v_{i_{2}},\ldots ,v_{i_{k}}]$$

For the mask performed on session s, which hides items with low importance scores, the positive view of the masking can be defined as follows:(14)$$s^{mask^{+}}=[v_{1},\ldots ,v_{i_{1}-1},[m],v_{i_{1}+1}\ldots ,v_{i_{k}-1},[m],v_{i_{k}+1},v_{n}]$$

GC4SRec is based on CL4SRec, where guided operations are used for contrastive learning to generate positive and negative examples. The loss function consists of two parts: recommendation loss and guided contrastive loss:(15)$$L=\sum_{u\in U}L_{rec}(s)+\lambda_{cl^{+}}L_{cl^{+}}(s)+\lambda_{cl^{-}}L_{cl^{-}}(s)$$
(16)$$L_{rec}(s)=-\log\frac{\exp(sim(h,h_{v^{*}}))}{\exp(sim(h,h_{v^{*}}))+\sum_{v^{-}\in V^{-}}\exp(sim(h,h_{v^{-}}))}$$
(17)$$L_{cl^{+}}(s)=-\log\frac{\exp(sim(h^{a_{i}},h^{a_{j}}))}{\exp(sim(h^{a_{i}},h^{a_{j}}))+\sum_{s^{a}\in S-S^{+}}\exp(sim(h^{a_{i}},h^{a}))}$$

h is the embedding representation of the session, s, $v^{*}$ represents the next item to be clicked in the session, and $v^{-}$ is the negative, $V^{-}=V-\{v^{*}\}$. $h^{a_{i}}$,$h^{a_{j}}$ are the contrastive views generated by mask and crop on the sequence, represented as $s^{a_{i}}$, $s^{a_{j}}$$S^{+}=\{s^{a_{i}},s^{a_{j}}\}$.
(18)$$L_{cl^{-}}(s)=-\frac{1}{\left|S^{-}\right|-1}\sum_{s^{a}\in S^{-}-\{s^{a}\}}\log\frac{\exp(sim(h^{a_{i}^{-}},h^{a}))}{\sum_{s^{a}\in S^{+}\cup \{s^{a}\}}\exp(sim(h^{a_{i}^{-}},h))}$$

## 4. Experiment

### 4.1. Experiment Settings

For all models with learnable embedding layers, the embedding dimension is set to 64, and the training batch size is set to 256. For each baseline model, all other hyper-parameters are set according to the optimal performance reported in the original paper. For GC4SRec, all parameters are initialized using a truncated normal distribution at [−0.01, 0.01]. The parameters are optimized using the Adam optimizer with a learning rate of 0.001. We tune the hyper-parameters range, which is set from 0.1 to 0.9, with a step size of 0.1.

### 4.2. Dataset

The first dataset is the MovieLens-1M dataset, which is widely used for evaluating recommendation algorithms. We conducted experiments on three public datasets collected from the real-world platforms. Two of them were obtained from Amazon, one of the largest e-commercial platforms in the world. They were split by top-level product categories in amazon. 

For dataset preprocessing, we coverted all numeric ratings or presence of a review to “1” and others to “0”. It is worth mentioning that to guarantee each user/item had enough interactions, we only kept the “5-core” datasets. We discarded users and items with fewer than 5 interaction records iteratively. The dataset information is shown in Table 1:

### 4.3. Evaluation

For each user, the last interacted item was used as test data, and the items before the last interacted item were used as validation data. The model ranked all items that the user has not interacted with according to their similarity. We employed hit ratio (HR) and normalized discounted cumulative gain (NDCG), which are widely used in related works, to evaluate the performance of each method. HR focuses on positive cases, while NDCG further considers ranking information. In this work, we report HR and NDCG with K = 5, 10.

### 4.4. Baselines

In order to prove the effectiveness of the model, GC4SRec was compared with six recommendation algorithms on three datasets. The characteristics of the six recommendation algorithms are introduced as follows:

GRU4Rec [1]: Uses gated networks to address the gradient problem in long-term memory and backpropagation.

GC-SAN [14]: Dynamically constructs graph structures of sequences and captures local dependencies through graph neural networks.

SASRec [2]: Uses the self-attention mechanism to model the user interaction sequence to capture the user’s dynamic interest.

S3Rec(MIP) [3]: Uses the self-supervised learning method to derive the intrinsic correlation of data and uses item masking data augmentation methods to solve data sparsity.

CL4Rec [4]: Uses a contrastive learning framework to derive self-supervised signals from raw user behavior sequences. It can extract more meaningful user patterns and further encode user representations.

DuoRec [5]: Proposes a dropout-based model-level augmentation and selects sequences of the same target item as hard positive samples.

### 4.5. Overall Performance Comparison

For each baseline, the embedding size was set to 64, and all other hyperparameter settings were kept the same as in the CL4SRec. The experimental results are shown in Table 2:

The metrics of GEC4SRec consistently outperform the latest algorithms on all datasets. GC4SRec outperforms DuoRec by an average of 2.8% over all metrics on all datasets, and by 28% over CL4SRec. The above results show that self-supervised learning combined with GNN guidance can achieve better performance. For contrastive learning, higher quality positive views lead to better representations of user behavior sequences. Experiments also show that the former is better than BERT4Rec, which fully proves that contrastive learning can effectively alleviate the data sparsity and improve recommendation performance.

GC-SAN can achieve better performance in some cases, proving the effectiveness of applying GNN in sequential recommendation. GC4SRec captures structural information in user behavior sequences by applying GNN and also considers the sequential information of user sequences. Therefore, better performance can be achieved.

## 5. Ablation Study

### 5.1. Necessity of Graph Neural Network-Guided Approach

To verify the effect of the GC4SRec graph neural network bootstrap on model performance, SC4SRec uses the attention mechanism to generate importance scores. CL4SRec removes the process of generating importance scores from the graph neural network compared to GC4SRec. The results are shown in Figure 4.

Figure 4 shows the evaluation results. It is clear that the bootstrapping process generates accurate contrast learning views, which have higher performance compared to CL4SRec, which generates contrast learning views randomly. The graph neural network has better results than the self-attentive mechanism in generating item importance scores because the graph neural network captures the structural information of user behavior sequences.

### 5.2. Necessity of Data Augmentation Methods

The common data augmentation methods for contrastive learning include mask and crop. To verify the impact of these two data augmentation methods on the model performance, experiments were conducted on three datasets with NDCG@5, and the experimental results are shown in Figure 5:

GC4SRec^mask-^ refers to the removal of the mask data augmentation method from GC4SRec, and GC4SRec^crop-^ refers to the removal of the crop data augmentation method from GC4SRec. On all three datasets, removing either of the data augmentation methods resulted in a decrease in recommendation metrics, indicating that both data augmentation methods can improve recommendation performance.

### 5.3. Study of Hyper-Parameter μ

During data augmentation, we choose to crop or mask the k items with the lowest scores, and k is defined by |μ⋅|s||, 0<μ≤1. The hyper-parameter μ determines the number of items with the highest score in the positive view under the bootstrap method. We employ the grid search to tune the hyper-parameter μ. In the experiments, the values were set from 0.1 to 0.9 for validation, and the results are shown in Figure 6 and Figure 7:

From the Figure 6 and Figure 7, it is clear the value of μ greatly affects the performance of the model. In the Beauty dataset, the optimal value of μ is 0.3. In the Sports dataset, the model’s performance is optimal when μ is 0.2. Therefore, different values should be used for different datasets to achieve optimal performance.

## 6. Conclusions

This paper investigates how to generate higher quality comparison views in sequential recommendations using a bootstrapping approach. We propose graph neural network-guided contrastive learning for sequential recommendations. We use graph neural networks to generate importance scores of items and construct contrastive learned views based on the importance scores. Experiments were conducted on three datasets, and the experimental results show that the model can effectively improve recommendation performance. Our work is the first to apply graph neural networks to address the problem of inconsistent positive views. In future work, we will investigate extracting the global contextual representation of users on the global graph and combining the global context and the sequence context to generate the user representation for item recommendation for users. The global context contains more global information, which can enrich the user representation, generate more accurate recommendation views for users, and improve the recommendation performance.

## Figures and Tables

**Figure 1 sensors-23-05572-f001:**
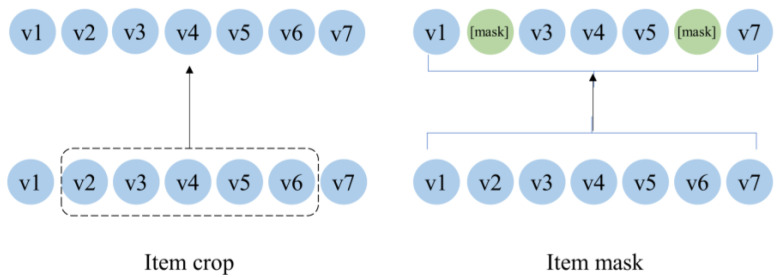
Augmentation methods.

**Figure 2 sensors-23-05572-f002:**
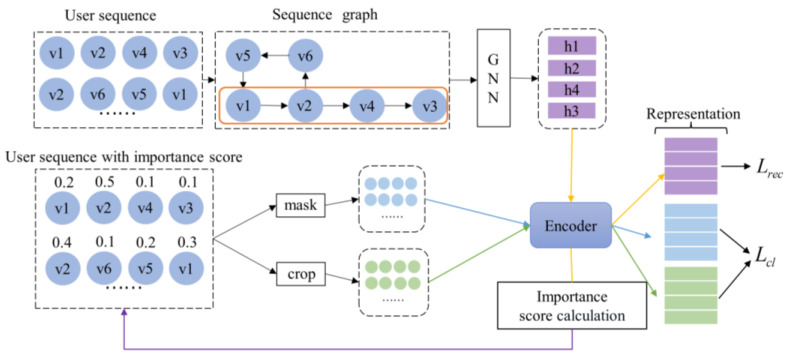
The framework of the model.

**Figure 3 sensors-23-05572-f003:**
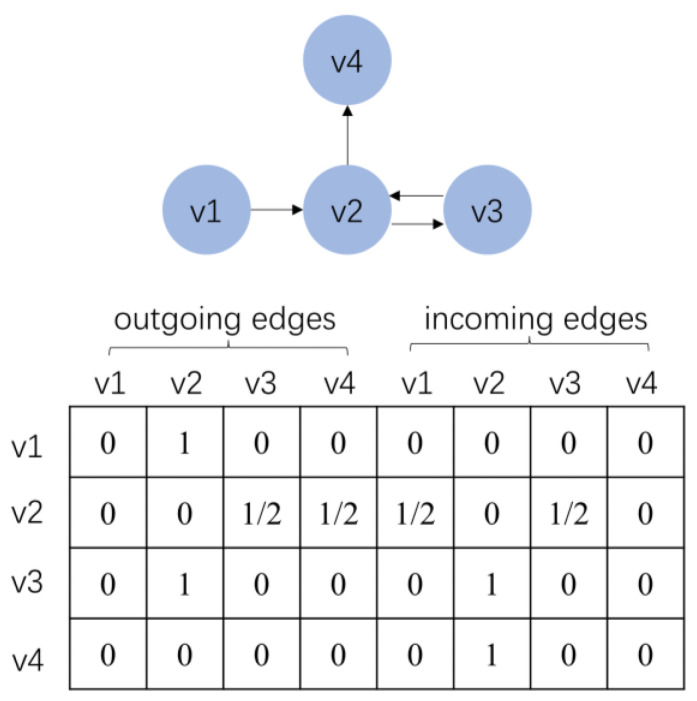
Sessions of a session graph and the connection matrix.

**Figure 4 sensors-23-05572-f004:**
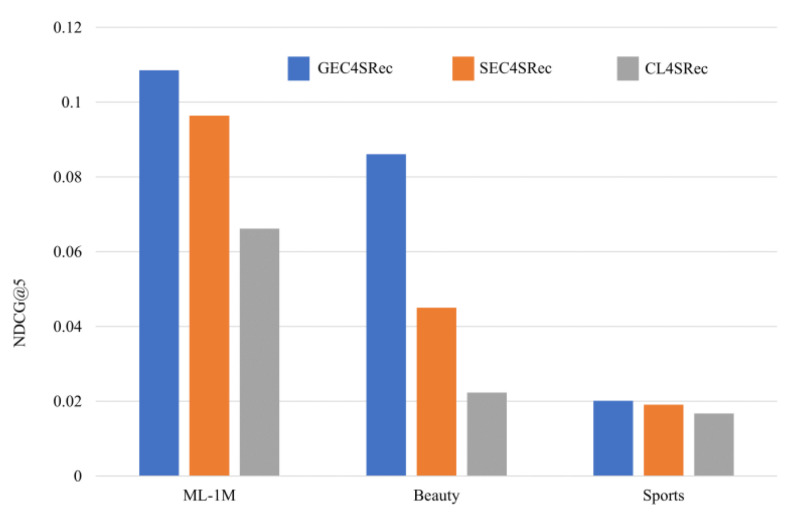
Ablation experiments with the necessity of a guided method.

**Figure 5 sensors-23-05572-f005:**
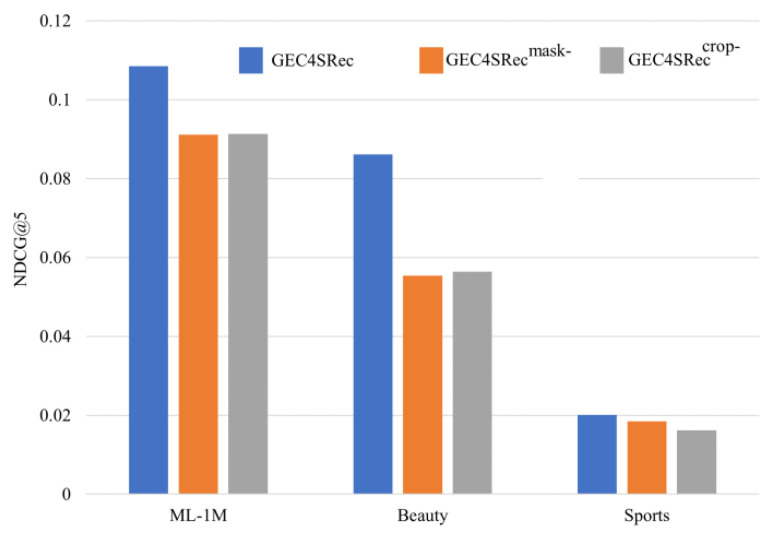
Ablation experiments with data augmentation methods.

**Figure 6 sensors-23-05572-f006:**
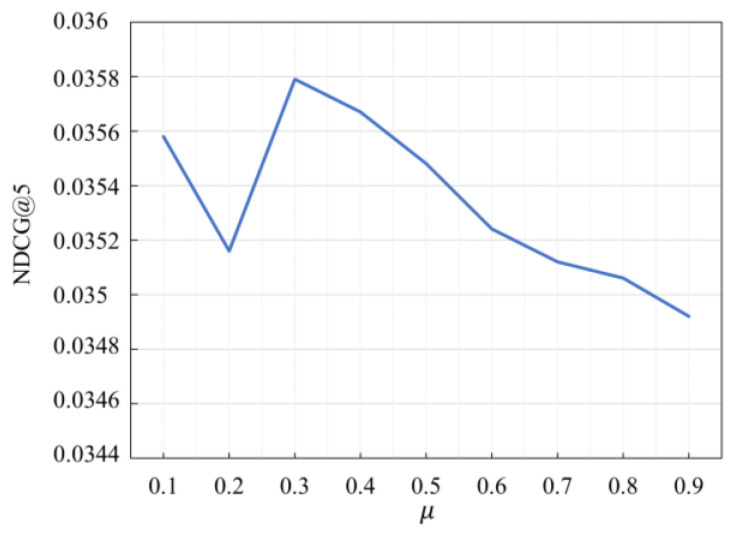
NDCG@5 with different μ settings on Beauty dataset.

**Figure 7 sensors-23-05572-f007:**
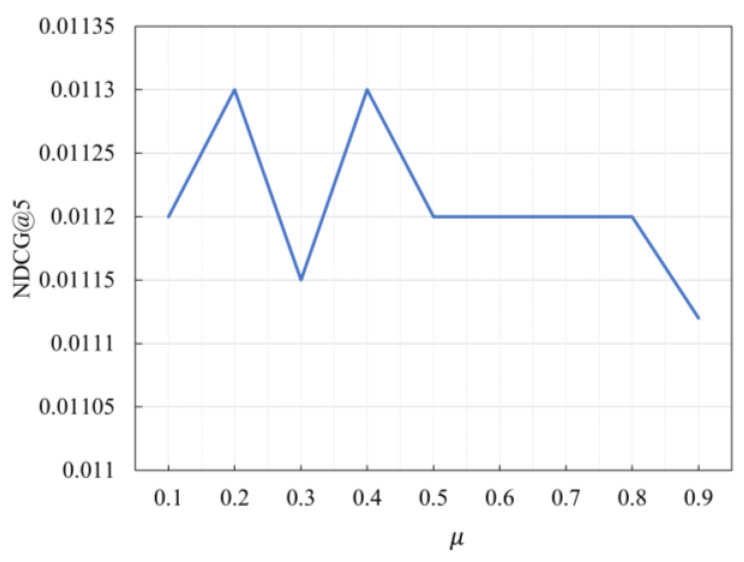
NDCG@5 with different μ settings on Sports dataset.

**Table 1 sensors-23-05572-t001:** Datasets.

Dataset	Users	Items	Actions	Density
ML-1M	6040	3953	1,000,209	4.2%
Beauty	22,363	12,101	198,502	0.07%
Sports	35,598	18,357	296,337	0.05%

**Table 2 sensors-23-05572-t002:** Experimental result on datasets for different models.

Dataset	Metric	GRU4Rec	GC-SAN	SASRec	S3Rec(MLP)	CL4Rec	DuoRec	GEC4SRec
ML-1M	HR@5	0.0763	0.01185	0.1087	0.1078	0.1147	0.1672	0.1685
	HR@10	0.1658	0.1862	0.1904	0.1952	0.1975	0.2507	0.2547
	NDCG@5	0.0385	0.0654	0.0638	0.0616	0.0662	0.1076	0.1085
	NDCG@10	0.0671	0.0925	0.0910	0.0917	0.0928	0.1345	0.1493
Beauty	HR@5	0.0164	0.0270	0.0365	0.0327	0.0401	0.0548	0.0588
	HR@10	0.0365	0.0444	0.0627	0.0591	0.0683	0.0832	0.0836
	NDCG@5	0.0086	0.0169	0.0236	0.0175	0.0223	0.0345	0.0648
	NDCG@10	0.0142	0.0225	0.0281	0.0268	0.0317	0.0436	0.0553
Sports	HR@5	0.0137	0.0174	0.0218	0.0157	0.0277	0.0310	0.0311
	HR@10	0.0274	0.0266	0.0336	0.0265	0.0455	0.0480	0.0482
	NDCG@5	0.0096	0.0110	0.0127	0.0098	0.0167	0.0191	0.0201
	NDCG@10	0.0137	0.0139	0.0169	0.0135	0.0224	0.0246	0.0485

## Data Availability

This study did not report any data. We used public data for research.

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
