# Peer review of "Graph Neural Network-Guided Contrastive Learning for Sequential Recommendation"

_sensors, 2023, doi:10.3390/s23125572_

Round 1
Reviewer 1 Report
The paper discusses the use of graph neural networks and contrastive learning in sequential recommendation systems and it provides an overview of the state of the art in sequential recommendation and contrastive learning, and then presents a new approach that uses graph neural networks to capture local context information and an encoder to extract sequence context information to generate importance scores. The paper provides a detailed description of the proposed approach and its advantages over existing methods. Overall, the paper is well-written and provides useful insights into the use of graph neural networks and contrastive learning in sequential recommendation systems. In addition, the description provides a clear and concise summary of the model and its components.

Reviewer 2 Report
The author propose Graph Neural Network guided Contrastive learning for Sequential Recommendation which can can enhance recommendation performance and mitigate the data sparsity problem.
This article still needs some modifications
1. English writing needs improvement, such as uppercase and lowercase letters(line 9, We. line 9,18,45,48,Graph Neural Networks)
2. Image quality needs to be improved(Fig.1. When you use a PDF reader, you will notice that the purple and green vertical lines are skewed.)
3. When the abbreviation GNN first appears, its full name needs to be marked.
4. line 115 (The font of t needs to be modified.)
5. In the conclusion, you need to introduce the importance and influence of your research.
English writing needs improvement.
1. uppercase and lowercase letters(line 9, We. line 9,18,45,48,Graph Neural Networks)
2. Try not to use too long sentences (line 303~307)
Reviewer 3 Report
In section 3.1, S is defined to be a set and represented as [s1, s2, ..., sn], while sequence s is represented as {v1, v2, ..., vn}. However, the notation {e1, ..., ex} is the standard mathematical notation for sets, while ordered tuples are typically represented as (e1, ..., ey). The authors should adopt the standard notations to avoid confusion.
In section 3.1, the authors describe a process for converting the sequence of interactions to a graph. However, in a graph two vertices are either connected with a single edge or not connected, while in the sequence of interactions a particular ordered pair of interactions may appear multiple times. The authors do not describe how this aspect is handled; in the experimental section, the authors state that "they discard duplicated interactions", however this is not sufficient because
(a) it appears at some distant part of the text which occurs later in the manuscript,
(b) is not detailed enough, in the sense that it does not accurately describe the granularity or the mode of "duplicates" detection. For instance, are "duplicates" considered at item-click level (if item X has been clicked multiple times, only one is retained), at item pair level (if item sequence (Xi, Xj) occurs multiple times, only one is retained, or at another level? In all cases, it is unclear which occurrence is retained, e.g. the first, the last, some random one?
c) dropping duplicates leads to loss of information since duplicate interactions may convey strong preference of the user to specific items or item sequences; the authors should discuss the implications of their choice to drop duplicates.
The authors refer to weighted connections between items, i.e. the edges of the graph are weighted. This is illustrated in Fig. 2. However the authors do not describe how weights of edges are computed. From Fig 2 it may be assumed that the weight in the input edge area is computed as 1/(#incoming edges to target node), however this is a speculation; the authors should explicitly list how weights are computed and discuss any implications of the weight computation approach.
Equations 12 and 13 define the same quantity scrop+, listing however different definitions. This should be fixed.
The alignment of equations 12, 15, 16, 17 needs rectification.
Equation numbers (19) and (20) appear stray, with no equation associated with them.
In section 4, the authors state "We tune the hyper-parameters range which is set from 0.1 to 0.9,with a step size of 0.1."; if this implies the use of grid search to tune the hyper-parameters, this should be stated. In all cases, the optimal hyperparameter values determined for each experiment should be listed in the manuscript.
Typesetting of mathematical notations needs to be improved. Especially mathematical notations occurring within the regular text flow are rendered in many cases to be disproportionally larger than normal text.
In Figure 1, LCL uses a larger font size than Lrec; probably this is unintentional and should be corrected.
English is fine, proofreading is advised to correct minor issues. For instance the text
S = [s1, s2 ,......, sn ] , st denotes the set
on line 115 should read
S = [s1, s2 ,......, sn ] , where st denotes the set
On line 160, score is misspelled as socre (5 instances).
